# Experiences and Expectations of the Characteristics of Friendly Primary Health Services from the Perspective of Men: A Phenomenological Qualitative Study

**DOI:** 10.3390/ijerph191912428

**Published:** 2022-09-29

**Authors:** Muhammad Zikri Ab Aziz, Tengku Alina Tengku Ismail, Mohd Ismail Ibrahim, Najib Majdi Yaacob, Zakiah Mohd Said

**Affiliations:** 1School of Medical Sciences, Health Campus, Universiti Sains Malaysia, Kota Bharu 16150, Malaysia; 2Family Health Development Division, Ministry of Health Malaysia, Putrajaya 62590, Malaysia

**Keywords:** experience, expectation, men, friendly primary health services, qualitative study

## Abstract

Inadequate men’s engagement with health services may be influenced by unmet needs and demands of the local men’s community. This study aimed to explore men’s experiences with primary health services and their expectations of the characteristics of friendly primary health services, from the perspective of men in Kelantan, Malaysia. A qualitative study using in-depth interviews was conducted with 15 men from six primary health facilities in Kelantan, Malaysia, who were selected based on maximum variation sampling. The data were transcribed and analyzed using the thematic analysis method. The study found that experiences with the existing primary health services were categorized into four subthemes: provision of health services, health promotion delivery, attributes of healthcare providers, and the physical environment of the health facilities. Meanwhile, the expectations of the characteristics of friendly primary health services were categorized into four subthemes: meeting the needs of men in primary health services, approaching men through effective health promotion strategies, standards of a healthcare provider from the viewpoint of men, and a comfortable physical environment for men. Prior experiences hugely influenced men’s expectations of friendly primary health services. Men want these health service characteristics customized to meet their needs, allowing them to use health services with confidence and comfort. Thus, to strengthen primary health services for men, it is essential to comprehend their prior experiences with and expectations of the services.

## 1. Introduction

Men’s health is a field that promotes men’s physical, mental, and social well-being throughout their life cycle and addresses health problems related to men [1]. The concept of men’s health concerns a broad range of issues, not just specific diseases of male reproductive organs [2]. In 1999, the New South Wales Health Department defined men’s health as any problem, condition, or factor that impacts men’s quality of life and/or for which alternative solutions are needed for men (and boys) to experience the best possible social, emotional, and physical health [3].

Globally, men have shorter lives than women, and, in 2016, the life expectancy of men at birth was 4.4 years lower than that of women [4]. In Malaysia, the life expectancy of men is five years lower than that of women [5]. In Kelantan, the gap in life expectancy between men and women is especially high, at 6.6 years [6]. Kelantan is one of 13 states in Malaysia and is located on the East Coast of Peninsular Malaysia. The population consists of Malays (93.8%), Chinese (3.0%), Indian (0.3%), other ethnicities (0.6%), and non-Malaysians (2.3%) [6,7]. The estimated population is 1.91 million, with men accounting for 0.96 million [6]. The agricultural industry accounts for the main economic activity in Kelantan, primarily fishing, rubber tapping, and paddy planting [8].

Despite the advancement of the population’s health overall, men’s health still lags, as evidenced by men’s shorter life expectancy and higher morbidity and mortality compared to women [1]. Furthermore, men have a higher risk of premature mortality than women, and men suffer from more life-threatening chronic diseases, such as heart disease, stroke, cancer, atherosclerosis, and kidney disease [9]. Several factors lead to this health disparity among men, including men’s attitudes, risk exposure, health-seeking patterns, and responses from the health system [4]. In addition, men are prone to involvement in more dangerous and destructive habits, such as smoking, alcohol abuse, fighting, and illicit drug use, compared to women [9,10].

Men are less likely to seek assistance with health issues [11], which translates into inadequate health service attendance [4]. In addition, some men have reported dissatisfaction with their experiences when they received treatment in the clinic [12]. This problem is thought to be related to cultural stereotypes, which influence men and may cause them to ignore screening and preventive healthcare and delay help-seeking for symptoms [13]. Moreover, researchers believe that men delay their use of preventive health services because of traditional masculinity social constructs, including extreme self-reliance and stoicism relating to health [14]. Most previous discussions on men’s help-seeking have positioned men as hesitant consumers or “behaving badly”. However, the design and quality of health services, including health providers’ attitudes and practices toward male clients, may also contribute to men’s low engagement with healthcare and are often underestimated [12]. Therefore, the low engagement of men in health screening and other health services is influenced by many factors, including the fact that healthcare is not tailored to their needs [15].

Primary healthcare in Malaysia is provided through a two-tiered system: health clinics and community clinics. The services provided by health clinics have improved from outpatient, maternal, and child health services to the concept of a family doctor and personalized care, to build a patient–doctor relationship between each family and dedicated doctors, covering primary level curative as well as promotional, preventive, and rehabilitative components of primary healthcare services. Meanwhile, community clinics play a role in ongoing maternal and child healthcare and basic curative services [16]. The infrastructure of health clinics in Malaysia differs based on the type of clinic and the estimated population in the coverage area. In addition, the estimated population covered by different clinics differs based on the clinic’s location, ranging from 10,000 people in rural settings to 100,000 in urban settings [16].

Differences in the efficiency of health services provision are observed between clients or patients of different genders and may result from how the services are built and/or delivered [17]. In addition, men may respond differently to healthcare services and health promotion messages than women, and there are differences among men regarding age, social, and ethnic groups [18]. Some men negatively judge health professionals’ role in health services, based on their previous unsatisfactory experiences and perceptions, which prevents them from seeking early treatment and health screenings at health facilities [19].

A recent study conducted in Kelantan on treatment-seeking behavior among a group of men suggested that to improve men’s health, health service policymakers and providers need to accommodate men’s needs rather than directing men to adapt to the existing healthcare system [20]. Moreover, there is a lack of understanding regarding local men’s expectations of the characteristics of friendly health services, due to inadequate data and studies. The World Health Organization (WHO)’s European strategy for men’s health, 2018, recommended improving the understanding of men’s health needs in order to improve future health services’ delivery to men [4].

Thus, an exploratory study is needed to understand the expectations of friendly primary health services characteristics from local men’s perspectives. Accordingly, this study aimed to explore experiences with primary health services and expectations of the characteristics of friendly primary health services from the perspective of men in Kelantan, Malaysia. This study is expected to add to the current knowledge on how primary health services should be designed to provide services that are friendly to men.

## 2. Materials and Methods

### 2.1. Study Design and Participants

A qualitative study using in-depth interviews was conducted from April 2021 to April 2022 at six government-based primary health facilities in Kelantan, Malaysia. Qualitative research helps understand men’s thoughts and feelings [21], which might influence their expectations of how friendly primary health services should be. In addition, a phenomenological approach was used to understand men’s expectations of the characteristics of friendly primary health services based on their perspectives, ideas, and experiences [21].

Adult Malaysian males aged 18 years and above who had previously attended primary health services were eligible to be involved in this study. The exclusion criteria were individuals who could not communicate in the Malay language. The sample size in the qualitative study was not calculated, but the estimated number was guided by reports from other studies. The final sample size was based on information saturation once the data collection and analysis started [22]. The estimated sample size was guided by previous studies on a similar topic, which had recruited approximately 14 to 17 participants to reach saturation [15,19,23].

The selection of participants was based on maximum variation sampling in terms of age, education level, income group, race, and residential area. The selection was made at selected primary health facilities, to ensure that only men who had experience with the services were involved in the study. The selection continued until data saturation was achieved based on the thematic analysis findings, when researchers no longer receive information from participants to add to the development of themes and subthemes related to the study.

Purposive sampling was conducted to select the primary health facilities and the participants. Two primary health facilities were purposely selected from Kota Bharu, Bachok, and Kuala Krai. These three districts represent different geographical areas in Kelantan.

### 2.2. Research Tool and Data Collection

An interview guide was developed for this study, consisting of open-ended questions assessing two main components: (1) experiences with the existing primary health services and (2) expectations of friendly primary health services from men’s perspectives. The open-ended questions were used to initiate the interview and as grand tour questions. In addition, follow-up and probing questions on more specific aspects were used to obtain more detailed information. The grand tour and probing questions are presented in Table 1. The open-ended questions approach was selected to stimulate discussion and included probes to address different dimensions. The questions were piloted and refined prior to and throughout the study.

The Kelantan State Health Department was approached and informed regarding the study. The researcher then informed the person in charge (PIC) of the selected primary health facilities. Potential participants were identified and invited to participate in an in-depth interview on a voluntary basis. A matrix of participants’ characteristics was created to ensure a diverse sample and control the balance of the participants. The interview was conducted face to face in a location convenient for the participants and arranged based on the participants’ preferences. They were informed of the study protocol and asked to provide informed consent.

The participants completed a proforma on sociodemographic details before the interview. All the in-depth interviews were conducted by a single researcher in the Malay language and were audio-recorded. They started with open-ended grand tour questions, followed by probes and follow-up questions, to allow participants to build on their original statements. When the conversation was silent, words that did not bias the direction of the conversation were used to restart the conversation. To prevent bias in the conversation, the researcher avoided sentences directed toward issues of interest, including signaling. Further, the researcher’s body language, expressions, and words were kept neutral. Each interview lasted from 30 to 50 min.

A memo was written including observational statements and non-verbal details during the data collection. The general feeling about the interview and the nuanced memorable conversation were also recorded to provide an accurate and meaningful reflective guide later. The interview was transcribed verbatim and analyzed before conducting another interview with a different participant. The data collection and analysis were conducted concurrently, until data saturation was observed.

### 2.3. Data Analysis

The thematic analysis method was used in this study. It analyzes qualitative data to identify patterns or themes within the data and uses them to address the research objectives [24]. There are six steps or processes of thematic analysis, which include familiarization with the data, generating initial codes, searching for themes, reviewing themes, defining themes, and writing up the results.

The first step was reading and re-reading the transcripts to become familiar with the data. The primary researcher, who initially conducted the interviews, read the transcripts, made notes, and jotted down early impressions. The data were then organized in a meaningful and systematic way, and coding was done to reduce the bulk data into small chunks of meaning. The coding involved each segment of data that was relevant to or captured something interesting about the research questions. An open-coding technique was used, as the codes were developed and modified during the coding process. The identified codes were discussed among the research team members, and revisions were made before moving on to the rest of the transcripts.

The codes were organized into subthemes and themes that reflected something specific about the research questions. The subthemes and themes were defined, examined for coherence, and ensured to be distinct from each other. All the codes, subthemes, and themes were continuously discussed with all research team members, and improvements were made after every data-collection episode. Data saturation was reached after interviewing 13 participants, and two additional interviews were conducted to ensure and confirm that no new themes or subthemes emerged. Thus, the final subthemes and themes were obtained from a total of 15 participants.

## 3. Results

Maximum variation sampling of participants from various sociodemographic backgrounds was performed, as shown in Table 2. Most participants were aged 35 years and above, with only one participant below the age of 35 (31 years old). The majority were from the Malay ethnic group and were married. Eight out of fifteen participants had attained tertiary education, and more were from the private sector compared to government servants or unemployed groups. Most came from the lowest income group, and 11 had medical comorbidities.

There were two main themes related to the study objectives: (1) experiences with the existing primary health services and (2) expectations of the characteristics of friendly primary health services from a men’s perspective. These two main themes emerged from four subthemes, respectively. The themes and subthemes are summarized in Table 3, followed by a detailed description of the findings in subsequent sections.

### 3.1. Experiences with the Existing Primary Health Services

All participants contributed to this theme’s findings. Four subthemes were related to experiences with the existing primary health services: the provision of health services, health promotion delivery, the attributes of healthcare providers, and the physical environment of the health facilities. Generally, the participants were satisfied with the primary health services at the local primary health facilities, although five of them expressed some difficulties or problems while receiving the health services. Those findings are further explained in the sections on those subthemes.

#### 3.1.1. Provision of Health Services

Three codes were grouped under the subtheme of provision of services: the waiting hour, quality of services, and medication supply.

##### Waiting Hours

According to the participants, efficient administration should result in short service waiting periods. Thirteen of them had experienced long waiting hours while using primary health services. They felt bored and distressed by the long waiting hours. They even expressed concern that they still had to wait for a long period even though there were only a few patients in the clinic, as illustrated by the participants’ quotes below:

*“I see there are not many people, 2 to 3 patients only* [waiting]*, after half an hour, and sometimes more than an hour; I still have to wait to be called. If I go (early) in the morning, the process is still not finished in the afternoon. It frequently happens to me.”*(R10, 47 years old, laborer)

*“When I go to a government health clinic, I go there in the morning, and yet in the afternoon* [I am] *still not finished. I do not know why; maybe too many patients.”*(R5, 60 years old, retired)

Two of the participants also believed that the long waiting hours could be caused by the doctors performing detailed history-taking and clinical assessments. Therefore, they sometimes avoided the government health clinics and went to the pharmacy or private clinics to avoid the long waiting hours. Most of the participants did not experience a problem with the pharmacy unit; however, a few said they had to wait for a long time at the pharmacy because there were many patients there. Additionally, they discussed their unsatisfactory experiences with emergency services in the health clinics, including the slow service given to the patients.

One participant, R5, a 60-year-old retiree from a private company, described inequality in the efficiency of the services offered in government and private clinics, noting that private primary health clinics managed their busy days better than government-based primary health clinics. Four of the participants also addressed the long waiting hours while at the registration unit. Non-systematic health card storage was believed to be one of the causes of long waiting hours, including R10, a laborer from Kuala Krai, Kelantan:

*“Out of the blue, my health card was lost* [in the clinic card storage]*. They will ask us back; when was the last time I came? For what* [reason]*? Then, the process of seeing doctors will be delayed due to [the] missing health card.”*(R10, 47 years old, laborer)

##### Quality of Services

Most participants did not question the clinical management of the healthcare providers, as they felt confident with their clinical knowledge and skills (*n* = 14). However, two of them viewed men as more interested in and trusting of traditional medication or unqualified health practitioners compared to medical treatment, including for common chronic illnesses and sexual health problems. They also expressed that some men preferred to buy medications at a pharmacy instead of getting medical advice first. This could also be caused by a lack of information on health services, causing men to find alternatives, as stated by R5:

*“There is no place to inform* [about sexual health problems or issues]*. Whom will people find? Consultant, the unqualified consultant who sells a product.”*(R5, 60 years old, retiree)

Regarding the quality of services provided, three participants mentioned that their expectations for adequate consultation time with the doctors might not be realized, especially during the busy hours with many patients, as the doctors tended to take brief histories. They felt disappointed with the short consultations despite the long waiting time. Further, they also reported inconsistent service quality in different primary health facilities and inadequate laboratory services compared to the hospital.

##### Medication Supply

Almost all of the participants had no problems with the medication supply, indicating that they were satisfied with the service and that the medications given were sufficient. However, one participant, R5, mentioned that he was given different types of medications for similar health complaints, which were not as effective as the medication given on the previous visit to primary health services. He was not given an explanation of why different medications were given, as illustrated in the following quote:

*“Since a few years ago, I* [have] *occasionally had back pain. I noticed that with similar health complaints, different clinics may provide different types of medications, and I was not informed why I have been given different medication that for me* [are] *not as effective as* [the] *previous medication.”*(R5, 60 years old, retired)

Another participant, R9, a 70-year-old man from Kuala Krai, Kelantan, complained about the difficulty of continuing medications at a different clinic, since he was a frequent traveler. He was given medication for one month for his chronic illness, despite being prescribed for three months, and it needed to be retaken monthly. Since he usually travels to another district in Kelantan, he found it challenging to continue the follow-up medication despite having a valid prescription. He mentioned that the other clinic required that he be reinvestigated at the clinic in order to get the medication.

#### 3.1.2. Health Promotion Delivery

Two codes formed the subtheme of health promotion delivery: awareness and acceptance of men. These codes are discussed below.

##### Awareness

The participants expressed that men might only seek treatment if they were experiencing a significant illness or sickness. They also agreed that men have poor health-seeking behavior, with some men prioritizing work over health, as indicated in the following quotes:

*“For me, only when I feel severely ill will I go. If had a normal fever, maybe tomorrow* [I will go]*. If not,* [maybe the] *day after tomorrow; if bad, I will go.”*(R14, 56 years old, fisherman)


*“The problem is men, indeed. I know because I am a man. Only if we had it severe, at home or other places, collapsed, carried, then we will go.”*
(R12, 57 years old, government servant)

Most of them knew that the primary health facilities had done various health promotions at the community level via outreach programs, but they were unaware of the promoted men’s health program (*n* = 8). For example, participant R1, a 39-year-old man, mentioned that he thought many older men have erectile dysfunction but do not know how to get help and delay seeking treatment. In addition, the majority of the participants were unsure about the availability of the services, or they thought that no services were available, since no promotion was seen. This is illustrated by R15, an older Indian man from Kuala Krai, Kelantan:

*“There is none* [sexual health services]*, there is none. That might be the reason why men do not want to go.”*(R15, 55 years old, lorry driver)

Most of them agreed that there was a lack or no promotion of men’s sexual health services, as illustrated by a few of the participants’ quotes:

*“So far, I have not experienced* [promotion]*. What I saw, there is health promotion of female *(sexual and reproductive)* health services like cervical cancer. For men, I do not think so, but I am not sure if I did not notice of that.”*(R10, 47 years old, laborer)

*“It is not available, why? Because we, our culture, our people will not talk* [about sexual health]… *like a man with hemorrhoids (health issue involving perineum area), they will not show it unless there is no other way* [due to severe bleeding or pain]*.”*(R11, 65 years old, pensioner)

*“At the clinic, I had not seen any* [promotion on men’s health]*.”*(R12, 57 years old, government servant)

##### Acceptance

The participants also mentioned that men are less likely to follow or join health programs compared to women. Regarding the current health promotion activities, they said that men might not engage well with the program. Specifically, men might not join the health promotion activities unless they are targeted specifically at their social group, as indicated by the following quotes:

*“The program done was good enough, good enough, depends on* [the] *acceptance of men themselves.”*(R9, 70 years old, pensioner)

*“When the health program is done in a community hall, no one gathered there, maybe 3 to 4 persons go there. If* [you] *want to attract more, you can do it by collaborating with local small religious groups, for example.”*(R11, 65 years old, pensioner)

Some current health promotion materials focus on the risky behavior of men, such as health education on the smoking ban, but men are usually not interested in the negative consequences of smoking cigarettes. R1 addressed this issue:

*“In general, negative health campaigns, such as on smoking habits, maybe he* [smoker] *will just take a quick glimpse at the advertisement and not read through because he already knows. It does not attract him, either to get treatment or to read the poster.”*(R1, 39 years old, government servant)

#### 3.1.3. Attributes of Healthcare Providers

The subtheme attributes of healthcare providers emerged from three codes found in the study: attitudes and behaviors, communications, and favoring male providers.

##### Attitudes and Behavior

Most of the participants were satisfied with the attitudes of the healthcare providers (*n* = 8). However, some participants noted that sometimes the attitude of a healthcare provider toward certain patients was inappropriate (*n* = 6). For example, the participants noticed that healthcare providers called patients using a loud voice and appeared to be in a bad mood. One participant, R4, a 45-year-old man from Bachok, Kelantan, reported that he had seen a patient being called with a loud voice and thought it could make the patient feel sicker and was inappropriate for the relationship between healthcare providers and patients. The situation could also lead to an unpleasant perception on the part of other patients witnessing the situation. In addition, a less caring attitude by the healthcare providers when answering patient questions could cause patients to feel disappointed. The following quotes reflect the situation some experienced:

*“Sometimes *(staff are)* in a bad mood when calling* [patient to enter the room]*, ah, yes, I had experienced hearing a healthcare provider call a patient in* [an] *angry/stress tone, in Kelantanese described as ‘mechok’, for example, “Hey! Come here! Please hurry! Meaning with a little stress.”*(R12, 57 years old, government servant)

*“I had experienced, in a place, they are like, do not care much, do not come* [to give early treatment] *although the patient had a wound and* [was] *bleeding.”*(R15, 55 years old, lorry driver)

The participants also mentioned that healthcare providers could sometimes be unfriendly to men, especially older adults. This view was expressed by four participants. For example, based on his previous experience, R2, a 59-year-old government pensioner who lived in the district of Kota Bharu, Kelantan, recalled that he saw a healthcare provider scolding an older man who walked slowly into the room. He thought the older man might have a hearing problem, causing him not to notice the healthcare provider’s call. The quotes from R2 are presented below:

*“Fierce, not friendly, because this man* (patient)*, sometimes they are old people, they walk slow, have poor hearing. When he was called once, he does not hear it, right? The staff was angry and raised their voices, and I have noticed that.”*(R2, 59 years old, pensioner)

*“When the healthcare provider calls elderly men, the elderly might not hear well or sit far away,* [so] *the patient* [does] *not arrive as soon as possible and is scolded by the healthcare provider like this: ‘Where did you go? Don’t you hear me calling?’* (in a loud and rude tone)*.”*(R2, 59 years old, pensioner)

However, not all of the participants were concerned with this issue, as they understood that the nature of work might place the healthcare providers in stressful conditions, so sometimes they might unintentionally use a loud voice with the patients, as illustrated in this quote from R8, a man from Kuala Krai, Kelantan:

*“When work involves* (with) *stress, we should already understand, understand* (the situation). *We just ignored it; sometimes they might get stressed…* (they are) *not angry, but their voice, high tone, haha.”*(R8, 65 years old, retired)

The majority of the participants acknowledged that healthcare providers were friendly to the patients, but a few thought that such friendliness might depend on patients’ behavior toward them. For example, R14, a 56-year-old fisherman from Kota Bharu, Kelantan, stated that the healthcare providers’ behaviors and attitudes might depend on how patients behave; if patients talk nicely with them, they will be friendly. Moreover, the participants noted that friendliness could depend on the individual and sometimes varied in distinct clinics, as mentioned below:

*“*[At the] *counter, the initial part, we take a number, usually* [looked like] *egoist… not sure* [how to explain]*,* [looked] *like they do not want to entertain, ah, as we just pay RM 1* [one Ringgit Malaysia] *at the clinic.”*(R6, 42 years old, company manager)

*“Ha! Less friendly, but at the children’s* [health unit] *a little different, more friendly, but in general* [at the] *health unit the services are lacking* [in friendliness]*.”*(R6, 42 years old, company manager)

However, not all of the participants had a similar experience, such as R6. In addition, a few other participants mentioned that they were served well by the registration staff and other support staff. One participant, R3, mentioned that friendliness could vary by occasion or time, as he noticed the same healthcare provider showing a different attitude at different times. Some of the participants questioned the dedication of certain healthcare providers, reporting that they had experienced situations where healthcare providers refused to hear more about what they wanted to say due to the need to do other things. R10 added that sometimes he witnessed the doctor arrive late to the clinic, causing the consultation session to start late. According to R4, one possible reason for these problems was an unprofessional attitude on the part of the healthcare providers:

*“Maybe the healthcare provider had a problem* [at home]*, then he brought it to the workplace. Then, other people* [patients] *become the victims.”*(R4, 45 years old, store assistant)

In addition, the participants felt some healthcare providers were not dedicated to work and wasted time on other things (e.g., chit-chatting), rather than doing the job quickly. This is illustrated by the following quotes from R5 and R6:

*“Sometimes they are chit-chatting with each other,* [but] *the patient’s document was not yet taken* [to be sent to doctor’s room]*. After* [they] *finish chit-chatting, they take the health card. If they take it early,* [I] *suppose the patient will be called earlier.”*(R5, 60 years old, retiree)


*“There are many patients, but the healthcare providers chit-chat a lot, among them… chit-chatting, sometimes in the triage area, they are chit-chatting with each other while treating the patient.”*
(R6, 42 years old, company manager)

##### Communication

Communication between healthcare providers and patients in the clinics happens as soon as patients enter the clinic for the registration process, consultation, investigation, medication supply, and other procedures. The participants described their experiences in communication-related matters, mainly related to how they occurred and the barriers to communication. Most of them were happy communicating with doctors and pharmacists, as illustrated by quotes from a few participants below:

*“The doctor’s* [explanation] *is ok, and the pharmacist’s explanation is also ok.”*(R6, 42 years old, company manager)


*“So far, I understand what the doctor explained. So far, no problem.”*
(R10, 47 years old, laborer)

One participant, R11, a 65-year-old pensioner from Kuala Krai, Kelantan, expressed his gratitude for the services the registration staff provided, as he believed they did their best to ensure smooth registration for all patients, and if a problem occurred, they would apologize and inform him of what happened. This showed good communication between the healthcare providers and clients (patients). However, another participant, R4, had an opposite experience of poor communication with the registration staff, as his number was not called after a few hours, although the next patient had already been called, and he was not informed why. This reflected a barrier in communication between healthcare providers and clients. R5 described the situation as follows:


*“There was a time I went early, but my name was not called. I was confused as a patient after me had already been called. I think I came first, after about two hours, then my name was called, and I was not informed why my turn was late.”*
(R4, 45 years old, store assistant)

Poor communication leads to patient having unpleasant views of the health service provided by the health facilities. Some of the participants experienced poor communication with healthcare providers, especially when asked about long waiting hours and being told to wait.

Regarding the explanation of the clinical investigation, some indicated that they were not informed of the findings. One participant, R5, expressed disappointment that he was not informed of the results and raised the possibility of a gap between healthcare providers and patients, leading to poor communication. He perceived a barrier to communicating with the healthcare providers, as shown in his quotes below:

*“I do not know the findings of my urine sample. The same goes for blood* [investigation finding]*.”*(R5, 60 years old, retiree)

*“I hope healthcare providers will inform* [about the results]*, but I am afraid to ask.”*(R5, 60 years old, retiree)

*“Between we* [patients] *and them* [healthcare providers]*, there are many gaps, differences. We are not from their *[level]*. We are just normal people; we have different thoughts. Their thought is different. When we communicate with them, we do not want to talk like we know more than them.* [I] *suppose they know more.”*(R5, 60 years old, retiree)

##### Favoring Male Providers

Almost all participants had no problem discussing general health issues with doctors of any gender, but they felt uncomfortable discussing sexual and reproductive health issues with female doctors. They felt embarrassed to talk about sexual health with a female doctor and might not give full details of the problem or even refuse to talk, as discussed in the following quotes:

*“Regarding getting treatment related to men’s problem or problem-related to male genitalia, itchiness, or other related symptoms, maybe* [I] *feel shy* [to inform a] *female doctor.”*(R1, 39 years old, government servant)

*“If possible, male* [healthcare provider] *for male* [patients]*, female* [healthcare provider] *for female* [patients]*, so if men want to seek treatment, he will not feel little shy.”*(R3, 31 years old, technician)

#### 3.1.4. Physical Environment of the Health Facilities

In addition, the participants reported good experiences with the physical condition of the primary health facilities, when their needs and expectations were met. As a result, a few codes emerged and were grouped under the health facility physical environment subtheme, which include cheerfulness, comfortability of the waiting area, the consultation room, and visitor-friendly amenities.

##### Cheerfulness

Most of the participants accepted the general cheerfulness of the clinics and were not too concerned about this matter, as illustrated by R6 and R7:


*“I think cheerfulness of the clinic is acceptable for men.”*
(R6, 42 years old, company manager)


*“For me, it is ok. As far as I look at this clinic, it seems cheerful.”*
(R7, 39 years old, clerk)

However, R5 had a different perspective, indicating that the environment of the current health clinic was not cheerful enough. He said that the clinic felt like being in a mortuary and was not colorful, which could improve the atmosphere.

##### Comfortability of the Waiting Area

Four of the participants expressed that they were uncomfortable in waiting areas with limited seats. Among those who were satisfied, they described the character of a comfortable waiting room as being large enough, having a functioning television, being located near toilets, having adequate chairs, and featuring exciting reading materials. In addition, they added that a functioning television and exciting reading materials helped to reduce their boredom while waiting.

With the advancement of technology, most adult patients have phones, but according to the participants, they could not fully utilize their phones while waiting, if the phone coverage was not good and there was no free Wi-Fi for patients. In addition, a few of them raised concerns about the comfort of a mixed waiting area for both men and women, and sometimes the presence of children playing could cause an uncomfortable situation for men, as described by R4 and R5:

*“Uncomfortable! Because it* [waiting area] *is mixed for men and women. The majority who sit are women. Men* [who have a seat] *can be counted* [small number]*. Rarely* [found]*. The majority* [of men] *wait outside the clinic. We felt very uncomfortable.”*(R4, 45 years old, store assistant)

*“Uncomfortable because… number one* [reason] *when, we, men, only two males* [are] *in the clinic, others are women, chit-chatting among them. We felt stressed.* [Also]*, a child making noise/crying, we felt like, just get out and back home. It happened to me. After I had registered, I turned back.”*(R5, 60 years old, retired)

An insufficient number of chairs in the waiting room usually caused men to stand while waiting outside the clinic, as priority was given to women, children, and the elderly to use the chairs, making men feel discriminated against based on the social norm.

##### Consultation Room

More than half of the participants (*n* = 8) felt unsatisfied with the condition of the consultation room, as they were uncomfortable with the small and shared room, while the rest were satisfied with the current condition. One participant, R7, a 39-year-old man from Kota Bharu, Kelantan, mentioned that he had a satisfying experience in the consultation room/area when it was large and was not shared with another patient. However, most of the other participants expressed dissatisfaction, mainly due to a lack of privacy, when they shared the room with other patients during a consultation. Some did not accept sharing a consultation room for any health issue, but others tolerated sharing a room when dealing with non-sexual health issues.

*“The feeling while sharing the room to get this treatment is that we do not feel privacy to express or to know the problem because there are other people* [in the room]*… maybe we have to speak a quietly so that other people do not hear what our problem is.”*(R1, 39 years old, government servant)

A few mentioned that small consultation rooms made them uncomfortable, including R5:

*“The area* [consultation room] *is narrow. Sometimes another patient sits next to me. It is too near* [between patients in a consultation room]*, and when we enter the room, I feel like* [I am] *in a tight situation.”*(R5, 60 years old, retired)

He also added that he had experienced disappointment when he did not feel welcome to have a good conversation with the doctor, since there was signage at the consultation door warning patients not to take a long time in the consultation room to prevent others from waiting too long. He described the situation as follows:

*“I do not want to ask or talk much because it stated outside* [at the door] *that the clinic is in a hurry because more people are waiting. On the outside* [of the door]*, there is a picture/signage warning* [us] *not to take a long time* [in the consultation room]*.”*(R5, 60 years old, retiree)

##### Visitor-Friendly Amenities

Most of the participants were not satisfied with the shortage of parking spaces and needed to find parking far from the clinics (*n* = 10). Sometimes, they had to park in the wrong parking space because there was no parking area near the clinic. Certain patients might require a parking area near the clinic due to an inability to walk or the severity of their illness. The geography of parking areas that were not on the same level as the clinic could also be problematic for certain patients:

*“The problem is the parking problem, lack of parking space. The reason why people do not want to go* [to the clinic] *is due to* [the need] *to park far from the clinic.”*(R12, 57 years old, government servant)

Two of the participants also worried about being cited by local authorities if they parked in the wrong place. Sometimes, they canceled their plan to get treatment at the health clinic because there was no parking available. However, for those who rode a motorcycle, parking spaces might not be an issue.

They had different experiences with the toilets in the clinics. Some of the participants had a poor experience with an inadequate number of toilets and had to share with another gender. For example, R3 shared that he needed to use the same toilet as female patients for urine collection, and he felt very uncomfortable with the situation. Other participants also had an unsatisfactory experience using the clinic’s toilet because it did not function well, was poorly maintained, or was smelly. However, most of the participants indicated that the toilets in the clinics seemed to be in satisfactory condition, adequate in numbers, not smelly, and clean.

### 3.2. Expectations of the Characteristics of Friendly Primary Health Services from Men’s Perspective

Four subthemes were related to the expectations of characteristics of friendly primary health services from a men’s perspective: (1) meeting men’s needs in primary health services; (2) approaching men through effective health promotion strategies; (3) standards of a healthcare provider from a men’s viewpoint; and (4) a comfortable physical environment for men.

#### 3.2.1. Meeting Men’s Needs in Primary Health Services

The participants described several characteristics they expected of friendly primary health services to meet their needs. These characteristics were coded and grouped under this subtheme. The codes include efficiency, clear motto, monitoring, appropriate waiting time, an extension of the time for services, general health services for men, emergency services, and men’s sexual health services.

##### Efficiency

Men demanded that primary health clinics have systematic administrative management, with fast and proper patient health card distribution to the doctor’s room and modern technology to increase management efficiency. Modern technology is necessary today, as described by R2, who noted that even coffee shops have adapted technology to reducing waiting time and streamline procedures for preparing food and drink and sending it to the clients. Therefore, he suggested that primary health clinics should adapt similar technology to quicken their administrative processes. Moreover, the health services should be continuous and not interrupted due to a lack of staff, especially consultations with doctors.

Some participants highlighted that priority should be given to older or very ill patients, as they need simpler and faster services:

*“They need to categorize patients,* (old) *people with diabetes* (for example)*. These people need to go early, to enter* (the consultation room) *early.* [The have] *no need to wait among other normal people; normal people they are not very ill.”*(R5, 60 years old, retiree)

In addition, they also expected primary health services to improve based on demand, as people are now using them more for non-life-threatening conditions rather than going to the hospital. Some suggested that health clinics could provide a server or waitress in the waiting room to assist patients who require help. They believed that such providers could assist patients, give directions, and help the elderly in the clinic. However, not all of the participants agreed with this idea. One participant, R12, a 57-year-old government servant from Kuala Krai, Kelantan, expressed his disagreement with the need to add more staff as servers or waitresses:

*“In my opinion, the clinics already have many ‘waitress’. All* [healthcare providers] *are waitresses* [that can assist patients]*.”*(R12, 57 years old, government servant)

##### Clear Motto

Some participants mentioned the importance of the primary health clinics having a clear motto or objectives, including services related to men’s health. In addition, they wanted the clinic’s motto or objectives to be published, so they knew what health conditions are covered by the services, especially those they could benefit from. R15 offered his view on this:


*“The most important thing, what is the clinic for? For what? For all? Or for certain diseases only; that is important.”*
(R15, 55 years old, lorry driver)

##### Monitoring

Regarding improvement and evaluation of the health services, two participants suggested that a monitoring body should be present to communicate and receive feedback on the services provided. They should also provide clients feedback forms. They also suggested an accreditation standard for the health clinics to ensure efficiency and consistency in all clinics. Some suggested that the clinic should conduct regular morning briefings to boost staff morale and confidence, so they would be more enthusiastic about serving the community.

##### Appropriate Waiting Time

When visiting a primary health clinic, the participants were concerned about the amount of time they had to wait. Mostly, they wanted shorter waiting times, as they could not bear long waits and become bored easily. For some, a long waiting time is inappropriate for a sick person to endure. However, they felt a long waiting time is tolerable if they feel satisfied with the quality of the treatment received. Moreover, they were disappointed with what they thought was inadequate treatment despite waiting hours for a consultation. They felt that if the clinic could provide information on the estimated amount of time they would have to wait before receiving a consultation, it might be an added value. They could then estimate when it would be their turn and do other things while they waited, as noted by R2:

*“Usually, the doctor has their estimation* [on the time allotted for a consultation]*, about 10 min per patient, and if I go to the* [registration] *counter, they should* [be] *able to give* [the] *estimated time before my turn, for example, at 3:40 p.m., I should have met the doctor.”*(R2, 59 years old, pensioner)

##### Extension of Services Time

Extended service hours or times, which might include services in the late evening or on the weekend, could benefit men who might not be able to come during office hours due to job commitments, as mentioned by three of the participants. In addition, one participant, R14, observed that people could get sick anytime, even on the weekend. Thus, he suggested the clinic should also be operated during the weekend.

##### General Health Services for Men

The participants expect good services and practices from healthcare providers. Expertise in the primary health clinic, including a counselor and medical specialists, could attract men to the clinic. In addition, the population’s rapid growth has increased the demand for the country’s health services. The participants indicated that they expect a high standard quality for all primary health facilities and the ability to perform advanced laboratory investigations to reduce dependence on the hospital. They believed that men differed from women, as once men were unsatisfied with the services, they would not return to that clinic. Some also expected health service providers to provide a regular health program or checkup for men, similar to well-established health programs for women (*n* = 5). However, they mentioned that there was no regular checkup for men of any group except for those with chronic illnesses. They also mentioned that prevention programs for men were lacking:

*“Best if we make the screening, like* [as an example] *blood taking for every three months for patients, meaning they need to do it, but it is not available for those without illness, if possible.* [It is] *good to have regular* [health screenings]*.”*(R12, 57 years old, government servant)

*“Support for those without illness to come to the clinic to have health assessment* [or screening]*… now, if possible, young men also need to go to* [the] *clinic.”*(R5, 60 years old, retiree)

Regarding services provided by pharmacy units, three participants emphasized the importance of high-quality medications. For example, R15 mentioned that he noticed particular medications given by private clinics were more effective in reducing pain. Moreover, the supply of follow-up medications for chronic diseases should not be problematic for travelers when they have a valid medication prescription slip. R9 strongly emphasized this point:

*“Just continue the medication! So there is no need to do things* [repeating the lab investigation]*; some might like it, but others might not, with the procedures.”*(R9, 70 years old, pensioner)

##### Emergency Services

Three participants expressed their thoughts about the provision of emergency services in a primary health setting. They said that the availability of emergency services at primary health clinics gives men confidence that the clinic can provide the necessary health services, especially during emergencies. One participant, R14, indicated that particular pain or health conditions might be inappropriate for hospital emergency unit visits but should be treated at the clinic level. Emergency services at the clinic level should be provided promptly to patients with a medical emergency or who were involved in an accident:

*“Need for strict action and* [to] *be fast. Sometimes an individual come*[s] *due to skidding from a motorcycle. He comes to the clinic first. If* [he] *cannot be treated at* [the] *clinic level, the clinic should send* [him] *to* [the] *hospital… before that,* [he] *needs to be treated first, because he is in pain.”*(R15, 55 years old, lorry driver)

According to the participants, the provision of 24 h emergency service in primary health facilities is essential for men, as they tend to use the services more, so providing such care in primary health facilities could increase men’s engagement with the clinic. In addition, an ambulance should also be made available in every primary health clinic so patients can be sent to the hospital when necessary.

##### Men’s Sexual Health Services

Most participants mentioned that men’s health services, including male sexual health services, are vital for providing male-friendly primary health services. The main staff who manage the men’s sexual health clinic should be men, to reduce male patients’ embarrassment. The majority agreed that the clinic should be operated at certain times only or on an appointment basis, as that would ensure privacy, since not all men need the services (*n* = 11). This could be done weekly or monthly, according to the number of patients and the clinic’s capability. However, some suggested such care should be available along with general health services. Meanwhile, two participants suggested that men’s sexual health clinics or services should be rebranded with a different name to reduce the stigma for those who receive treatment. The failure to provide men’s health services or men’s clinical services could lead to adverse health behaviors, as indicated by R5:

*“This* [the man’s sexual health problem] *becomes like a secret* [taboo]*… while it should not be a secret and should be discussed with a doctor, but cannot. Because* [it is] *not available* [the service]*.”*(R5, 60 years old, retiree)

*“When* [there is] *no place to discuss, whom do men meet? ‘Consultant’, the unqualified ‘consultant’ who sells* [unregistered health] *products.”*(R5, 60 years old, retired)

#### 3.2.2. Approaching Men through Effective Health Promotion Strategies

The participants described various ways and methods for healthcare providers to improve health promotions and education for men. The participants’ views were coded and grouped under the following subthemes: content, location, styles, approaches, Internet and social media, local men’s community, and acknowledgment.

##### Content

Nine participants recommended that healthcare providers spread more information about men’s health services or male-friendly health services to attract men and give them confidence in the health services. Health promotion and education could emphasize common diseases among men or men’s reproduction and sexual health problems and the available treatments at primary health facilities. In addition, knowing that the services are available would improve men’s engagement with local clinics, as mentioned by R1:

*“I think that health provider*[s] *should make an announcement on diseases related to men and when and where they can get treatment so that they get the treatment and not only stay at home.”*(R1, 39 years old, government servant)

Further, two participants added that promoting an active lifestyle may also help men manage their health conditions. Therefore, a variety of health promotion content is expected to improve men’s health literacy.

##### Location

The participants viewed a variety of locations as appropriate for health promotion, including clinics, social media, and mass media. In addition, health promotion activities could be done in gathering areas frequented by men, including religious places, coffee shops, or restaurants. A few participants discussed how such gathering areas favored by men could be used in health promotion:

*“Can make at* [the] *local community mosque… some doctors like to do community activities, so can give a talk at the mosque.”*(R6, 42 years old, company manager)

*“Men like to hang out at the coffee shop, and many men are there, so* [they] *can do* [promotion] *there casually.”*(R4, 45 years old, store assistant)

Posting health flyers in coffee shops or restaurants may also reach the men who like chit-chatting there. According to some of the participants, the community hall is a good option as a location to educate men on men’s sexual health issues because it provides more privacy for men. On the other hand, open spaces, such as shopping complexes, are inappropriate for health education involving sexual health topics because they expose men to stigma, as mentioned by R9:

*“Supermarket, boring, because many people walk there. Men will walk and take a quick look… sometimes he will smile only, not stopping,* [thinking that] *whatever men* [are] *at the place must have that problem.”*(R9, 70 years old, pensioner)

##### Styles

Most of the participants expected polite, soft persuasion and the use of more positive words or approaches to promote health for men. For example, R1 talked about negative health campaigns, such as those related to smoking, arguing that men might not even read the health promotion materials, since they already knew about the negativity, and so it would not catch their attention. Another participant, R15, stated that men’s health promotion must be done using soft language or style:

*“Need to be soft, like say,* [focus on] *what* [is] *important for men… we need to treat men like that, tell them like that. Do not say in two to three months you could die.”*(R15, 55 years old, lorry driver)

Men must be encouraged and supported to change for better health and informed that health services are available to them. However, some of the participants thought that men needed a more direct form of health promotion to get their attention. For example, R6 indicated that threatening words might encourage men to get medical checkups at the clinic:


*“Show threatening words a little bit, then they will come. They will feel afraid and want to go to the clinic for a check-up.”*
(R6, 42 years, old company manager)

##### Approaches

At the clinic, participants expected the clinic manager to take the opportunity to promote health in the facility using various approaches, including giving short talks on health in the waiting area that could fill the client’s waiting time with a worthwhile message. In addition, discussion on health topics with the patients and accompanying relatives could improve patient engagement, thus promoting trust. R11 discussed this matter:

*“*[It is] *better to have a healthcare provider there, better a nurse… One staff, to give a talk, estimated about 20 min, 15 min.”*(R11, 65 years old, pensioner)

*“When many patients are waiting in the waiting area,* [give a] *talk, and* [be] *open for any question.”*(R11, 65 years old, pensioner)

Pamphlets could also be used in the clinics to educate men. Furthermore, the televisions that most clinics have in waiting rooms should be used to provide health education, as discussed in R6’s statement below:

*“In the clinic, we have* [a] *television, right? So make* [health] *programs* [or videos] *to be shown on the clinic’s television.”*(R6, 42 years old, company manager)

##### Internet and Social Media

Five participants expressed their opinions on using the Internet and social media to promote health among men. They mentioned that healthcare providers in the clinics should utilize the Internet and social media to engage with the local community. This could be done by regularly updating their activities and programs and providing a contact number for any queries. R3, who was working as a technician for a private company, discussed this issue:

*“The first thing is that* [they] *need to update the community, as nowadays people are more on the internet and use more social media, so the clinic should update the community. It needs to have its own social media account, so that it is easier for people to refer and see* [the available programs and activities] *and if they have any inquiries* [they can reach the healthcare providers]*.”*(R3, 31 years old, technician)

##### Local Men’s Community

Many participants believed that educating men’s communities requires healthcare providers to participate in community activities (*n* = 9). Men from different social groups or with different demographic characteristics might have different interests in community activities. Local clinic managers need to acknowledge this and plan community health activities accordingly. For example, men in Kelantan might be interested in Dikir Barat (traditional musical performance), sports such as football, cycling, and badminton, or karaoke. R14 discussed the added value of offering Dikir Barat as one of the side activities in a health program:

*“If we invite* [the community] *to join* [the] *health program just like that* [without interesting activities]*, maybe people will not come. To make it easy* [to understand]*, if we want to invite* [them]*,* [they] *need to include* [programs like] *Dikir Barat. Then, many people will come… most people in every district* [in Kelantan] *will like that; it is our cultural art.”*(R14, 56 years old, fisherman)

Clinic managers can plan to include these activities in their programs or join a venture community program by giving talks and offering health exhibitions. Moreover, health promotion activities could be done by collaborating with religious groups, as some men might engage well with religious programs or activities. One participant, R11, suggested that healthcare providers could engage with the local community where they live, by assisting the community in establishing and conducting health screenings among their community.

##### Acknowledgment

Two participants thought that promoting health among men must involve men as active participants. Men could be involved by giving them the responsibility to take care of their community. Quotes from two participants emphasized that giving men responsibility would make them feel more welcomed and appreciated:

*“They* [healthcare providers] *can attract regular patients to become the promoter of the clinic to attract men to go to clinic.”*(R2, 59 years old, pensioner)


*“We cannot call them directly to come without giving them responsibility. They will feel satisfied when given responsibility; they will feel satisfied.”*
(R9, 70 years old, pensioner)

#### 3.2.3. Standards of a Healthcare Provider from Men’s Viewpoint

In terms of their view on the characteristics of healthcare providers, the participants mentioned several key points they expect. The standards regarding healthcare providers’ characteristics are categorized under the following codes: dedication, admirable attitudes, good knowledge, professional work practice, preference for male healthcare providers, and older and experienced healthcare providers.

##### Dedication

Many participants said that being dedicated to serving patients is vital for gaining patients’ trust. The participants had various views on how healthcare providers could show dedication while working. These include being a good listener to patients’ complaints, talking nicely with patients, prioritizing patient care, being ready to serve, and showing care to the patients. In addition, being a good listener is crucial for health providers, especially doctors, as it greatly impacts their relationships with the patients. One participant, R15, emphasized that doctors are responsible for allocating time to listen to patients’ health complaints:

*“It is your responsibility* [the doctor]*. You become a doctor, we find you…. A person who wants to be a doctor, he must treat the patients well because the government pays your wages and takes care of you, so you need to take care of patients. People come, and you need to take care* [of them]*,* [you] *need to discuss* [communicate] *well.”*(R15, 55 years old, lorry driver)

##### Admirable Attitudes

Most of the participants believed that being friendly would strengthen relationships with the patients. A healthcare provider who always has a smile on their face, greets patients, and is approachable could attract men to the clinic, and men themselves would spread the information about the friendliness of the healthcare providers in the community, thereby improving public trust.

According to the participants, if inappropriate services are given to men, especially adult males or elderly males, men will refuse to return, leading to poor health-seeking behaviors among men. In addition, a poor working attitude, such as chit-chatting with other healthcare providers while serving or treating a patient, can negatively influence the view of the services by men and should be avoided.

##### Good Knowledge

Five participants highlighted the need for healthcare providers to have good health knowledge to treat patients. Regarding men’s health, the men requested that healthcare providers have exposure to and be able to detect and offer early treatment for men’s health issues, including sexual health issues. This is not only true for doctors but also for other healthcare providers involved with men’s health.


*“I think maybe staff also need to have exposure to problems related to men so that they can be able to help to give the best treatment for the disease.”*
(R1, 39 years old, government servant)

*“Necessary, necessary, meaning that from any aspect if we* [patients] *ask, he or she* [is] *able to answer. Although not a doctor, he or she is a nurse* [and] *needs to have knowledge.”*(R3, 31 years old, technician)

##### Professional Work Practice

The participants evaluated good healthcare providers by observing their practice. Some stated that they wanted healthcare providers, especially doctors, to be thorough in their clinical assessments and treatment, asking the right questions before making a diagnosis. They also emphasized the importance of empathy and proper treatment by healthcare providers. R15 discussed this matter:

*“They need to take care, as* [with] *their relative, ah! Do not treat the patient like it is not their concern. Do not be like that; do not let the health problem* [remain] *not treated, and they should take care like your relative, like your parents, siblings.”*(R15, 55 years old, lorry driver)

Further, they also expected the healthcare providers to answer any queries appropriately. For example, they should not simply ask the patient to wait when they are asked about the reason for the long waiting period, as mentioned by R4. In addition, healthcare providers must have good communication skills and communicate using appropriate language and words that are understood by the community. R5 expressed concern about poor communication between healthcare providers and clients and recommended improvement:

*“Between we* [patients] *and them* [healthcare providers]*, have many gaps, differences… our thinking and theirs are not same.* [We] *feel afraid to communicate with them… suppose they* [should] *try to understand patients’ concerns and give a proper explanation.”*(R5, 60 years old, retired)

##### Preference for Male Healthcare Providers

The participants expressed strong preferences for male healthcare providers when discussing sexual health issues, as men might be shy to talk with female doctors about such issues. A team of male staff should handle men’s sexual health services if circumstances permitted. The support staff does not necessarily have to be all male. Most participants did not expect to be treated by male doctors for general health issues (non-sexual related health issues), as mentioned in R1’s statement below:

*“I think, if common diseases like cough, runny nose, there is no problem if* [men see a] *female doctor. But if, for example, to get treatment for men’s problem or problem-related to genitalia, itchiness, or other* [complaint]*, may feel shy to discuss* [the problem] *with a female doctor.”*(R1, 39 years old, government servant)

From another point of view, men might be attracted to visit the clinic due to the presence of female healthcare providers due to their gentle approach, as mentioned by R15:

*“To attract men… the clinic that has more female healthcare providers, men must go there, not due to sensual, it is different, it is like the mind, mind becomes calm when men go there. The way female healthcare providers treat men, he will tell other men to go to the clinics due to* [the] *gentle services of the female healthcare providers.”*(R15, 55 years old, lorry driver)

##### Older and Experienced Healthcare Providers

The participants prefer to deal with aged healthcare providers, as they appear to have more experience and maintain patients’ confidentiality better than younger healthcare providers.

*“*[Men’s clinics] *need to handled by veteran staff, senior, that can keep secret, at the registration counter.”*(R2, 59 years, old pensioner)

However, most of the participants did not consider this a must, as long as the healthcare providers could treat them well and maintain integrity.

#### 3.2.4. Comfortable Physical Environment for Men

Due to budget constraints, some of the participants are aware that providing all the measures to ensure clients’ convenience is not easy. R11 highlighted this issue but emphasized that it would be helpful for patients’ satisfaction if the government could do so. In addition, the availability of primary health facilities near one’s residence is a factor influencing attendance. The participants’ expectations of comfortable physical environments are coded as followed: cheerfulness, placement of health support services, comfortable waiting area, men-only waiting area, privacy-protected and comfortable consultation room, specific men’s sexual health services setting, and visitor-friendly amenities.

##### Cheerfulness

Some of the participants discussed the cheerfulness of clinics (*n* = 9). They had different views, as some wanted colorful environments, so the clinic looked more “alive”, while others did not expect the clinic to be colorful, as it had little influence on men’s view of the services. The quotes below reflect the participants’ views on this matter:


*“It should be colorful, colors that enliven the atmosphere. This color can give good vibes to our soul.”*
(R5, 60 years old, retired)


*“I think cheerfulness of the clinic is acceptable for men.”*
(R6, 42 years old, company manager)


*“For me, it is ok. As far as I look at this clinic, it seems cheerful.”*
(R7, 39 years old, clerk)

##### Placement of Health Support Services

The placement of units in the clinic also needs to be correct to ensure that patients are not burdened or encounter difficulties. For example, R2 mentioned that a physiotherapy unit should not be located on the upper level, as patients with back pain may find it difficult to go there.

##### Comfortable Waiting Area

Participants expected clinic management to provide comfortable and adequate chairs for every client in the waiting area and to separate them by unit. In addition, they were concerned about the division of the waiting area, as clients from different units were using the same waiting rooms, which caused congestion. One participant suggested that clinic management could provide massage chairs in the clinic, but others did not view this as an essential aspect. Others indicated clinics could provide coffee-shop-like chairs and tables to make patients more comfortable. Different types of chairs might also be considered, as not all patients can sit on a regular clinic chair due to health problems.

However, different chair types are not crucial to some participants, who reported that they were satisfied with the current chairs. Moreover, while waiting, men can get hungry, and the participants suggested that the clinic could provide free cold water and coffee or sell soft drinks and light food for snacking. This could help men cope with the long waiting times before receiving services.

*“Clinics should have drinking water point or place for long waiting,* [to] *drink water.”*(R7, 39 years old, clerk)

*“*[I] *want to sit, drink something, chit-chat while waiting.”*(R13, 41 years old, government servant)

##### Men-Only Waiting Area

Some participants suggested that the clinic manager should provide a men-only waiting area (*n* = 5). If the circumstances permit, such a waiting area would make men feel more comfortable. R2 noted that men like to be among men and to talk to each other about current issues. Men sometimes get irritated with mixed waiting areas, as mentioned by R5:

*“Like say, females, children, should not sit together with men. Please sit in another place. Please do not scream,* [it] *feels irritating.”*(R5, 60 years old, retiree)

However, not all of the participants requested a men-only waiting area. They felt the current conditions were acceptable, and the separation of seats based on gender could be problematic to people who come with their families. The clinic manager should ensure that the waiting area offers health promotion materials compatible with men, including posters and pamphlets about diseases and graphic information about men’s health problems and treatment. In addition, other reading materials that are not health-related should be available. Other suggestions by men included toilets near the waiting area, free Wi-Fi or stable phone coverage, a functioning and operating television, and the use of a loudspeaker to announce each patient’s turn.

##### Privacy-Protected and Comfortable Consultation Room

From the participant’s perspective, the most crucial characteristic of a consultation room is maintaining privacy. Men expect that they will not have to share a room with another patient during their consultation, especially when they want to talk about sexual health issues. Sharing a consultation area makes men uncomfortable, and they do not have privacy to talk about their problems, especially regarding sexual health. R13, a 41-year-old Chinese man living in Kuala Krai, Kelantan, spoke about this issue:


*“Sometimes shared with other doctors, so how does it feel to share? Truly uncomfortable because we do not know the other person, so like me, a lot of privacy problems… afraid to share.”*
(R13, 41 years old, government servant)

In a limited setting where sharing is unavoidable, two doctors should not share one table or face each other, and they should use a curtain to create a barrier to ensure privacy. Men are also concerned about the friendliness of the room for disabled people to enter and easily receive services. The size of the room also needs to be large enough.

##### Specific Men’s Sexual Health Service Setting

According to most of the participants, men’s sexual health services need to be available in every primary health facility, to encourage men to discuss their sexual health issues (*n* = 13). The setting should be a specific room to ensure the patient’s privacy. Patients might feel embarrassed if other patients can hear their conversation with the doctor.


*“The room should be a specific room that can ensure the privacy of the patients so that they can express their sexual health problem without feeling shy.”*
(R1, 39 years old, government servant)

##### Visitor-Friendly Amenities

Regarding other areas in the clinic, the participants stated that the toilet should not be shared between genders and should be functioning and clean. In addition, a children’s playground area would be more convenient for men who come with their families, allowing their children to be entertained while they receive treatment.

Furthermore, they demanded that clinics provide more parking spaces, since, currently, most people use cars to go to the clinic. Inappropriate parking on the sidewalk could result in them being cited by local authorities. In addition, a lack of parking might cause men to cancel the clinic trip altogether. The participants suggested that clinic management could request outside land from the local authority for use as a parking area for clients and ensure that the parking area is not abused by someone parking there for days.

## 4. Discussion

In this study, most of our participants were Malays (*n* = 13), in line with the sociodemographic characteristics of the residents of Kelantan. In addition, most of the participants were middle aged or older (*n* = 14). The main reason for this is that more older adults were available in the clinics during data collection and were more cooperative in terms of sharing their experiences and expectations related to the study topic.

Creating a more patient-centered healthcare system that provides excellent responsiveness to patients’ needs, expectations, and preferences has become the main agenda for improving the provision of health services in recent decades [25]. To create a patient-centered health system, the stakeholders need to know how patients experience current services and their expectations. In 2014, Price et al., described patient experiences as any process that a patient can view, including subjective feelings (e.g., pain control), objective experiences (e.g., waiting more than 15 min past the appointment time), and observations of the behavior of the physician, nurse, or staff (e.g., the doctor provided all relevant information) [26]. A positive patient experience usually leads to higher-quality health care, including patients’ engagement and adherence to medical professionals’ instructions, a better clinical process, and improved outcomes.

The majority of participants were generally satisfied with the primary health services they received (*n* = 10), but they felt that there was room for improvement in terms of providing friendly services. They noted that there is still a lack of efficiency in management and a lack of services provided, specifically for men. Long waiting times also caused the participants to feel uneasy, as they admitted having low patience for waiting. A similar finding was reported in an earlier study on male civil servants in Kelantan, in which one of the reasons for poor treatment seeking by men was a lack of patience. Thus, the men preferred to buy medications at the pharmacy [20]. A systematic review of the barriers and facilitators of health screening in men found that, in the health-system domain, long waiting times and a few other issues related to efficiency could hinder men from accessing screening services [27]. This finding supports our finding that some issues related to efficiency may cause men to have poor experiences with health services.

The participants noted that men’s health promotion activities are lacking, which could cause men to have poor health literacy and to exhibit poor health-seeking behaviors. They were also not well aware of men’s health services (*n* = 7) compared to maternal and child health, which they viewed as well established. As men’s health is a relatively neglected subject compared to women’s health, men have been left behind in terms of health screenings and programs [1]. Poor promotion might be due to a lack of emphasis on men’s health programs at the primary care level. In addition, men also expected regular health programs, including regular health screenings and checkups. This finding is supported by a previous study showing that men complained about insufficient health services and requested routine health screenings and checkups, similar to those offered to women [28]. Thus, a clear motto or objective of men’s health should be promoted and visualized at the clinic, to make men aware of the services and make them feel confident in them.

Men are usually alert to the attitudes and behaviors of healthcare providers during their visits. Negative attitudes and behaviors on the part of healthcare providers, such as talking harshly with patients or not showing care for patients, influence men’s perceptions of healthcare providers. In turn, these negative perceptions affect the relationship and trust between providers and patients. According to Sweta Dcunha et al., patients rarely express a belief that physicians who show care for the patients will provide better care and do their best to treat them, though this is something they desire [29]. Non-professional acts during working, such as chit-chatting while serving patients, was another factor that can shape men’s negative perceptions of healthcare providers.

Most participants focused on their experiences in the waiting and consultation rooms when asked about their experiences with the physical conditions of the primary health facilities (*n* = 14). A crowded waiting room with limited seats causes an unpleasant feeling as they wait for their turn for services. Shared and small spaces for consultations also make patients feel uncomfortable. A literature review by Jazla Fadda on the impact of the physical environment on quality of care found that health facilities with a privacy-protected environment, adequate space, and some visitor-friendly amenities were expected to improve patient satisfaction [30]. This could be correlated with our study findings, as the physical environment clearly had an effect on patients’ experiences and satisfaction.

Regarding the participants’ expectations, we found that most of the participants related their experiences with their views and expectations on how the services should be provided (*n* = 14). A previous study conducted in South West Ethiopia showed a substantial correlation between patient satisfaction with waiting and consultation times, the types of investigations conducted, and securing the prescribed medications [31]. Our participants mentioned that long waiting times were the main problem that could hinder them from using the services, as they feel irritated and bored by long waiting times. Further, the expectation of a long waiting time might deter men from seeking health assistance [28]. Thus, the provision of fast services is something that men covet and would help to attract men to clinics.

Adapting modern technology that can improve management efficiency is also essential. Such technology could reduce the burden on staff as well as waiting times. For example, the use of online medical records could reduce the staff’s workload related to record keeping and retrieval processes. This also could aid in improving overall primary health services, which have been expanded to include coordination of care across multiple providers and the management of complex diseases. In addition, the use of technology in primary health facilities could increase physicians’ capacity to provide integrated, accessible care, bringing specialty consultations into primary care for patients [32].

Some of the participants addressed the inconsistency in effectiveness between clinics (*n* = 3), and they suggested an accreditation system to ensure high-quality health services. This finding is consistent with C. Reeve et al., who found that the evaluation of health service performance is essential in health services planning to improve service performance and community health outcomes [33]. In addition, extended service hours can give working men the opportunity to receive health services in government health clinics. Emergency medicine services should also be available in clinics, as the participants indicated that many men would benefit from such services. Some also expected emergency services to operate for extended hours or even 24 h (*n* = 3). A qualitative study on men’s health needs found that employed men report having time constraints in caring for their health, and they have difficulty accessing primary care services due to limited hours of operation [28].

Health promotion for men should be done in a positive way. They should not be blamed or labeled as having poor attitudes and behavior; rather, they should be encouraged and supported to improve their health and use the services provided. Men dislike a negative persuasive style and believe it to be ineffective. A previous study on mental health promotion among men showed that the use of “male-sensitive” language and activity-based approaches allowed for healthy emotional expression, social involvement, and open communication between men and health providers [34]. Conversely, harsh words or language might hurt men’s confidence in the quality of health services and reduce their engagement with health services. Likewise, health promotion or education that uses a threatening style might not be well-accepted by men.

Giving responsibility to men to be involved in men’s health promotion in the community might also improve the confidence and esteem of local men. The results of a previous study support this finding, showing that giving responsibility to an individual in the community, such as a community health champion role, can serve as a catalyst for change at the individual and community levels [35]. In addition, the relationship between men and providers can be improved if the clinic has a social media account and uses it to interact with the community. A study on patient preferences for social media utilization and content in infertility clinics found that most participants felt that social media improved patient experiences in the clinics, and most felt comfortable communicating electronically with the clinic [36]. Therefore, social media is a more practical way to engage with the community, especially the new generation.

The majority of the participants agreed that health clinics should provide men’s sexual health services in specific men’s clinic settings (*n* = 14). Such a setting is expected to include male doctors. In line with our findings, an earlier qualitative study exploring factors influencing health screening behavior in young men also found that Muslim men usually preferred doctors of the same gender, because they felt more at ease and found it easier to develop mutual understanding, especially when discussing sexual health issues or having a genital or rectal examination [37].

A previous study found that men’s perceptions of healthcare providers were influenced by masculinity, as men may not trust healthcare providers to be able to give good treatment, and some men felt they knew better than the providers [38]. In our study, several participants indicated that they trusted the doctors’ ability and knowledge of general health issues, but they expected health providers to have more knowledge of men’s sexual health services, including detection and treatment (*n* = 5). Thus, having experts in the clinic could improve men’s confidence and trust in the services. Furthermore, our study found that the vital characteristics of health providers from a men’s perspective include working professionally, showing empathy to patients, and being comfortable communicating, which is consistent with the findings of a previous study on men’s expectations of physicians in relation to sexual health concerns [38].

According to the participants, a friendly physical environment is characterized by comfortability, confidentiality, and convenience. An article published in 2014 in *Urology Times* listed three vital design elements, suggested by urologists and internists, when setting up a men’s health center: (1) an entrance, reception, and waiting area that is separated from other units in the clinic; (2) a masculine decor, which makes men feel at ease; and (3) plenty of sports, men’s magazines, televisions, and, if possible, free Wi-Fi [39]. In addition, almost all of the elements mentioned by the participants in our study, including toilets, should be properly maintained and functioning well. The participants also highlighted the limited parking spaces and said that clinic managers should provide more patient parking spaces. None of the participants mentioned the need for having a separate entry for men’s sexual health services in primary health services, as it would be impractical to have separate entries in existing primary health facilities.

Our study was an exploratory study involving three districts in Kelantan. The participants were selected based on maximum variation sampling to ensure a variety of socioeconomic backgrounds. This allowed various voices from different groups of men in Kelantan to emerge, sharing their experiences and expectations of the characteristics of friendly primary health services. Thus, the results offer a more precise account of what would make primary health services friendlier for men. In addition, the in-depth interviews allowed men to express their thoughts without bias from other participants, which resulted in rich data. All participants were patients or relatives of patients who come to the clinic for general primary health services.

Several limitations of the study should be mentioned. The COVID-19 pandemic limited the selection of participants, as some health services were unavailable during this time. Thus, this study did not directly include patients with recent experiences of using the men’s sexual health services provided by the primary health facilities. Although the combination of in-depth interviews and focus group discussions can be expected to improve the understanding of the phenomenon of interest [40], no focus group discussion was conducted due to the risk of transmission of COVID-19. The participants’ experience was not limited to the recent past (e.g., within five years). Instead, we allowed the men to express any experience they could recall. As significant improvements have been made to primary health services over time, an old experience might not relate to the current health services. However, older experiences might be used to understand the experiences men have had with health services that could affect their perceptions.

Future research is needed to establish men’s expectations of primary health services in a more extensive research population, to help local health authorities provide better health services for men. In addition, more research is needed on the provision of male-friendly primary health services, to encourage more men to use health services.

### Reflexivity and Study Rigor

As a new qualitative researcher, the primary researcher might have values and perspectives that influenced the study’s implementation, including the data analysis and interpretation. He has been a medical doctor for almost nine years and has attended extra courses on qualitative research at a university and elsewhere. He was aware that his academic background might play a part in the conduct and analysis of this study. As an adult man living in Kelantan, his background, beliefs, and experiences could affect any aspect of the research. Therefore, in conducting this study, he received guidance from experts in qualitative research, including the other research team members involved in qualitative studies and men’s health studies, to reduce the potential effect of researcher bias and to recognize and address any ethical issues. In addition, living in Kelantan might have helped him to communicate in the local dialect, which provided a comfortable medium for the participants to talk and discuss the issues. Although he is Malay, the researcher has experience dealing with patients with various backgrounds, including Chinese and Indian patients, and has been exposed to their cultures and world views.

The development of the conceptual framework of this study involved a thorough literature review to provide a logical and convincing argument for the research. The qualitative study’s sampling design was purposive sampling to ensure the inclusion of appropriate participants in the study. The data collection and analysis were performed concurrently, with the results of the ongoing analysis informing the data collection until data saturation was achieved. A matrix of participants’ characteristics was created to ensure a diverse sample, to control the balance of the participants, and to ensure informed consent before the interviews started. The confidentiality and privacy of the participants were protected to the highest degree possible.

Member checking was done to ensure the credibility and confirmability of the study, which involved providing the interview transcripts to the participants [22]. This was done to verify the completeness and accuracy of the transcripts and to ensure that they reflected the participants’ meanings and intent. Credibility was also enhanced through peer debriefing, in which the study’s codes, themes, and conclusions were presented to the research team members and a few men’s health experts. The feedback that was given helped to enhance the credibility and validity of the study.

Investigator triangulation was also applied in this study. The first and second authors reviewed all audio recordings and performed individual data analyses to identify patterns or themes related to the aims of the study. The findings of codes, subthemes, and themes were compared and discussed with all the research team members. Improvements were made, based on a series of discussions, until a consensus was reached. Validity was established once all the researchers reached the same conclusion. Investigator triangulation entails the involvement of two or more researchers in the same study to obtain multiple observations and conclusions [40] and is expected to add depth to the topic of interest by providing both confirmation of the findings and other viewpoints [41].

## 5. Conclusions

Engaging men with primary health services requires healthcare providers and managers to not only cater to men’s health needs but also consider their expectations regarding how the services should be provided. Thus, it is vital to understand the experiences of men using existing primary health services and their expectations of the characteristics of friendly primary health services, to design services that are friendlier to men. Our study found that men’s experiences with the existing primary health services could be described in four subthemes: provision of health services, health promotion delivery, attributes of healthcare providers, and the physical environment of the health facilities. Meanwhile, men’s expectations of the characteristics of friendly primary health services were elaborated in four subthemes: meeting the men’s needs in primary health services, approaching men through effective health promotion strategies, standards of a healthcare provider from a men’s viewpoint, and a comfortable physical environment for men.

## Figures and Tables

**Table 1 ijerph-19-12428-t001:** Interview guide.

No.	Interview Topic	Grand Tour Questions	Probing Questions
1	Experiences with primary health services	Can you tell me about your experiences using primary health services? How did you feel about using the services? Did you have an unforgettable experience while using the services?	How about your experience during the registration proses? How would you describe the treatment you received? What were your experiences dealing with healthcare providers? Have you had any positive or negative experiences with health care providers? Can you elaborate more? How would you describe health promotion?
2	Expectations of friendly primary health services	From your point of view, what are the characteristics of friendly primary health services? In your opinion, how should health services be friendly? Can you explain why?	From a men’s perspective, what are the physical characteristics of friendly primary health services? Can you explain why? What characteristics of healthcare providers do you expect in friendly primary health services, from a men’s perspective? Can you elaborate on that?

**Table 2 ijerph-19-12428-t002:** Characteristics of the participants (*n* = 15).

Variable		Total (*n* = 15)
Age (years)	Young adult (18–35)	1
	Adult (36–55)	7
	Senior (56 and above)	7
Ethnic group	Malay	13
	Chinese	1
	Indian	1
Marital status	Married	13
	Widower/divorced	1
	Single	1
Education level	Primary	1
	Secondary	6
	Tertiary	8
Employment status	Government employee	3
	Private sector worker	7
	Unemployed/pensioner	5
Household income	<RM 4850	9
	RM 4850–RM 10,959	5
	RM 10,960 and above	1
Chronic medical illness	No known medical illness	4
	Single medical illness	6
	Multiple medical illnesses	5

**Table 3 ijerph-19-12428-t003:** The themes and subthemes identified from the thematic analysis.

Subthemes	Themes
Provision of health services	Experiences with the existing primary health services
Health promotion delivery	
Attributes of healthcare providers	
The physical environment of the health facilities	
Meeting men’s needs in primary health services	Expectations of the characteristics of friendly primary health services from a men’s perspective
Approaching men through effective health promotion strategies	
Standards of a healthcare provider from a men’s viewpoint	
A comfortable physical environment for men	

## Data Availability

Not applicable.

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
