# Peer review of "Experiences and Expectations of the Characteristics of Friendly Primary Health Services from the Perspective of Men: A Phenomenological Qualitative Study"

_ijerph, 2022, doi:10.3390/ijerph191912428_

Round 1
Reviewer 1 Report
The authors need to be careful with using words such as the word poor as it has a bad connotation. When the authors say they did triangulation, what forms of data did they use? There is no indication of what these other forms of data were. In the methods section the authors do not indicate what is the sample size, only how saturation impacted the sample size. This information is in the data analysis section even though it should be in the participants' section.
The reflexivity section can go in the discussion section as it gives reasons why results were found.
For the themes and subthemes the authors may want to present the themes as a tree where the theme is at the top and below are the subthemes or use word clouds.
When presenting the results it is not important which respondent # said what, but what a specific respondent said.
"The participants described that an efficient administration should result in short service waiting periods" How many participants said this? Consider putting ( n = #). If it was all participants put All participants. The same goes when you say Some Participants or Most participants, what is the (n = #). Please make sure your results reflect this comment above to make the findings more clear.
Make sure in the discussion you clearly indicate how many participants you are talking about when you say Some participants, most participants.
The authors should be able to identify several future research topics instead of solely one. That section does not go in the conclusion, but in the discussion. The authors do not indicate any limitations to their study. Every study has limitations.
Reviewer 2 Report
The manuscript is about the views and experiences of men about primary health services in Malaysia. The subject can be interesting but unfortunately I think manuscript is not proper for publishing:
1- The English writing of the manuscript is very poor, and it needs massive editing. Even many Quotations are meaningless. As examples I bring some of them here:
- “The space is a little narrow, not wide, sometimes the seat next to this (place currently seated) is 586 too near, when we enter the door, feel like blocked”
- “waitresses are already many, haha, at the place, in the clinic already many, all (healthcare provid-669 ers) are waitress” R12
- “Usually, the doctor has their estimation (on time taken for consultation), about 10 minutes per 702 patient, like as a patient, if I go to the counter, able to tell directly, your turn is (estimated) at 3.40 703 pm. So we know, because the country of Turkiye already practices this, there is no need to wait 704 long. They already know that the counter can give an estimated time based on the number of pa-705 tients” R2
2- Manuscript doesn't provide any information about the health system in Malaysia, and the role of the clinics there, for example how many population do they cover, or if the physician is present all the time....
3- The main problem about this manuscript is its methodology and data analysis. The participants must include diverse men with different experiences, but almost all of them were married, Malay, middle aged men.
4- The data analysis was not appropriate, that this issue has been reflected in the result section. The result section is very long and the primary codes have not been clustered adequately. For example there are problems about the discussing sexual issues with the health personnel that can be seen in different themes and sub-themes. There are different sub-themes that could be categorized with each other, making the result section shorter and more helpful.
5- There are no novelty or interesting subject in the manuscript.
3-
